# Molecular Mechanisms of MmuPV1 E6 and E7 and Implications for Human Disease

**DOI:** 10.3390/v14102138

**Published:** 2022-09-28

**Authors:** James C. Romero-Masters, Paul F. Lambert, Karl Munger

**Affiliations:** 1McArdle Laboratory for Cancer Research, University of Wisconsin School of Medicine and Public Health, Madison, WI 53705, USA; 2Department of Developmental, Molecular and Chemical Biology, Tufts University School of Medicine, Boston, MA 02111, USA

**Keywords:** papillomavirus, HPV, MmuPV1, E6, E7, viral oncogenesis, animal models

## Abstract

Human papillomaviruses (HPVs) cause a substantial amount of human disease from benign disease such as warts to malignant cancers including cervical carcinoma, head and neck cancer, and non-melanoma skin cancer. Our ability to model HPV-induced malignant disease has been impeded by species specific barriers and pre-clinical animal models have been challenging to develop. The recent discovery of a murine papillomavirus, MmuPV1, that infects laboratory mice and causes the same range of malignancies caused by HPVs provides the papillomavirus field the opportunity to test mechanistic hypotheses in a genetically manipulatable laboratory animal species in the context of natural infections. The E6 and E7 proteins encoded by high-risk HPVs, which are the HPV genotypes associated with human cancers, are multifunctional proteins that contribute to HPV-induced cancers in multiple ways. In this review, we describe the known activities of the MmuPV1-encoded E6 and E7 proteins and how those activities relate to the activities of HPV E6 and E7 oncoproteins encoded by mucosal and cutaneous high-risk HPV genotypes.

## 1. Introduction

Human tumor viruses contribute to at least 15% of the world’s total cancer burden [1]. Research on RNA and DNA viruses that cause cancer in humans or in animal species have substantially contributed to our understanding of carcinogenesis including the discovery of many oncogenes and tumor suppressor genes, and our understanding of the molecular mechanism by which these genes contribute to cancer [2]. In particular, studies of transforming proteins encoded by DNA viruses, such as human papillomaviruses (HPVs), polyomaviruses, and adenoviruses, contributed extensively to our understanding of host-encoded tumor suppressor genes [3,4,5,6,7,8]. “High-risk” HPVs, those that cause cancer, contribute to approximately 5% of the world’s total cancer burden [1,9,10]. HPV-associated cancers include anogenital tract cancers such as cervical and anal cancers, a rising percentage of head and neck cancers, and non-melanoma skin cancers, the latter in patients with a genetic predisposition to epidermodysplasia verruciformis (EV) [1,9,10,11,12,13,14,15,16]. HPVs initially infect the proliferative, basal cells of a stratified squamous epithelium but viral genome synthesis and viral progeny synthesis is confined to terminally differentiated cells [12]. Hence, one of the key hallmarks of the papillomaviral replication strategy is to disrupt the link between keratinocyte differentiation and proliferation. As a consequence, HPVs primarily cause hyperplastic epithelial lesions, referred to as papillomas (warts) or dysplasias. Most of these HPV-induced lesions are transient in nature because host immune responses are thought to commonly resolve the infections. Nevertheless, HPV-induced skin warts are the most common dermatology disease treated today [17,18,19,20]. If an individual becomes persistently infected with one of the high-risk HPVs, there is an increased risk that the lesion will progress to cancer.

HPV can be classified into various genera based on their genome sequence [12]. Practically they can be distinguished by the type of epithelial tissues they infect. Mucosal HPVs infect the poorly keratinized epithelia of the anogenital tract and the oral cavity, whereas cutaneous HPV infect the highly keratinized epithelial skin tissues. The alpha genus mostly encompasses mucosal HPVs whereas the beta, gamma, and mu genera are mostly cutaneous HPVs [12]. Mucosal HPVs can be further classified as “high-risk” and “low-risk” HPVs based upon their etiological association with cancers [12]. Low-risk HPVs cause benign papillomas whereas the lesions caused by high-risk HPVs have a propensity to undergo malignant progression. In the case of cutaneous HPVs, certain beta HPV genotypes (e.g., HPVs 5 and 8) have been classified as high-risk HPVs because have been shown to be associated with the development of cutaneous squamous cell carcinomas in EV patients and shown to have transforming and carcinogenic potential in in vitro and in vivo models, respectively [21,22,23,24,25,26,27].

The high-risk, mucosal HPV E5, E6, and E7 genes encode proteins with oncogenic activities and have been extensively studied. They target cellular processes that have been linked to the hallmarks of cancer [28,29]. The cutaneous, beta and gamma HPVs lack E5 genes. The E6 and E7 proteins of some cutaneous HPVs have been demonstrated to display oncogenic activities in cell and transgenic animal-based studies [16,21,22,23,24,25,26,30,31,32,33,34,35]. Because of the pronounced species specificity of papillomaviruses, it has not been possible to perform pathogenesis studies by experimentally infecting laboratory animal with HPVs. While experiments with HPV transgenic animals have provided many valuable insights into the carcinogenic activities of individual viral proteins, their mechanisms of action and their synergy with host factors in causing cancer, these model systems have important limitations. Firstly, the HPV gene(s) are being expressed in epithelial tissues outside the context of a natural infection. Secondly, in most cases the mice are tolerized to the viral protein disallowing one from assessing the role of host immunity in HPV-driven neoplasia. The discovery of the murine papillomavirus, MmuPV1, that infects laboratory mice has provided us for the first time with an infection-based model experimental model in a genetically tractable laboratory animal host [36]. The utility of this model is being expanded and utilized to study PV-induced disease in ways that have not been achievable before. While this model is powerful, the understanding of the mechanisms by which the MmuPV1 E6 and E7 proteins contribute to viral pathogenesis remains limited. In this review, we will briefly discuss the activities of the HPV E6 and E7 proteins that contribute to pathogenesis, describe the current mechanistic knowledge of the MmuPV1 E6 and E7 proteins, and discuss how these activities may relate back to known activities of cancer-associated HPV genotypes.

## 2. Oncogenic Activities of High-Risk HPVs

High-risk mucosal HPV-induced disease is driven by the activities of three viral proteins, E5, E6, and E7. They play critical roles in the replication of HPV and their oncogenic activities reflect their viral replication functions. The high-risk” HPV E5, E6, and E7 oncogenes are well characterized and extensively studied. The high-risk HPV E5 protein activate the EGF signaling pathway [37]. Increased EGF signaling contributes to neoplasia by triggering PI3K and MAPK signaling pathways that promote cellular growth and inhibit apoptosis [37,38,39,40,41,42,43]. The high-risk HPV E6 and E7 inactivate the p53 and retinoblastoma (pRB) tumor suppressors, respectively [6,7,44]. HPV16 E6 also activates telomerase expression and activity, which, together with HPV E7 contributes to immortalization of human cells [45,46,47,48,49,50,51,52,53,54,55,56,57,58,59,60]. E6 and E7 are expressed in cervical cancer and are necessary for proliferation and survival of HPV-positive cervical carcinoma lines [61]. High-risk HPV E7 proteins interact with pRB through their LXCXE motifs and can trigger the degradation of pRB [44,62,63,64]. This abrogates pRB’s inhibition of E2F transcription factor activity [6]. Normally pRB’s interaction with E2F proteins is regulated by cell cycle dependent phosphorylation/dephosphorylation. High-risk HPV E7′s inactivation of pRB overrides the normal control of the cell cycle by pRB because it leads to persistent activation of E2Fs [6,44,65]. High-risk HPV E6 proteins target p53 for proteasomal degradation by binding the E6AP (UBE3A) ubiquitin ligase [66,67]. HPV E6-mediated p53 degradation protects the infected cells from p53-mediated antiproliferative and apoptotic cellular responses that are triggered in response to E7 expression. Several mechanisms have been postulated by which high-risk HPV E6 proteins also activate TERT, the protein subunit of the human telomerase that leads to the immortalization of cells in vitro [45,46,68,69,70,71,72,73]. Together with E6-mediated telomerase activation, the inactivation of these tumor suppressor pathways by the high-risk HPV E6 and E7 proteins are considered to be important mechanisms by which these viral oncogenes contribute to tumorigenesis.

In addition to inhibiting p53 and pRB, the high-risk HPV E6 and E7 proteins also contribute to viral pathogenesis through other mechanisms. Both the high-risk HPV E6 and E7 proteins have been implicated in the inhibition of the differentiation process [74]. Expression of the high-risk HPV E7 oncogene has been shown to elevate p63, a master regulator of basal cell identity, and in the case of HPV31 does so by decreasing the levels of miR-203 which is elevated during keratinocyte differentiation [75]. The mucosal HPV E7 proteins (both high- and low-risk) have been shown to degrade the non-receptor tyrosine phosphatase, PTPN14 (with HPV18 E7 having the strongest affect) through the E7-associated p600 (UBR4) ubiquitin ligase [76,77]. This E7 activity has been shown to inhibit initial steps in keratinocyte differentiation in 2D tissue culture-based models [76,77,78,79,80]. There is some evidence that the high-risk HPV E6 and E7 oncogenes alter TGF-β signaling to prevent keratinocyte differentiation with HPV16 E7 being shown to inhibit TGF-β induced repression of the MYC promoter [81,82]. However, the high-risk HPV E6 and E7 proteins have not been shown to directly target SMAD2/SMAD3 which are key transactivators of TGF-β signaling. Additionally, HPV16 E6′s degradation of p53 is a potential mechanism of preventing differentiation induced by NOTCH signaling as p53 has been linked to NOTCH-induced differentiation [66,67,74,83]. By causing the degradation of PTPN14, high-risk HPV E7 proteins increase HIPPO signaling thereby promoting basal cell identity [79]. High-risk HPV E6 proteins have been shown to increase YAP nuclear localization through its PDZ binding domain, potentially through interactions with the scribble (SCRIB) and angiomotin (AMOT) proteins. Hence, both E6 and E7 have been implicated in regulating pathways critical for maintaining basal cell identity [84]. The high-risk HPV E6 and E7 proteins have also been shown to reduce genomic stability by inhibiting p53, depletion of free nucleotides, activating replication stress, and by promoting mitotic aberrations [85]. Moreover, the high-risk HPV E6 and E7 proteins inhibit innate immune signaling pathways, including RNA and DNA sensing by the RIG-I and cGAS-STING pathways, respectively, and have also been reported to dampen adaptive immune response [86,87,88]. Collectively, all these activities likely contribute to how these viral proteins promote tumorigenesis in high-risk HPV-infected cells.

## 3. Biological Activities of E6 and E7 Proteins Encoded by Cutaneous HPVs

Cutaneous HPVs do not encode an E5 protein; however, cutaneous HPV E6 and E7 proteins have a number of pro-oncogenic activities including the ability of a subset of cutaneous β-HPV E6 and E7 oncoproteins to immortalize keratinocytes [26,30,89,90,91,92,93]. With the association of some the β-HPVs with non-melanoma skin cancers in EV patients and potentially also in immunosuppressed patients, a molecular understanding of the biological activities and the cellular targets of the cutaneous HPV E6 and E7 proteins is critical to our understanding how these viruses promote neoplastic skin disease [15,94,95,96,97]. The β-HPV E6 proteins have been shown to interact with MAML1, an essential co-activator of NOTCH signaling, and SMAD2/SMAD3, co-activators of TGF-β signaling, which are tumor suppressive signaling pathways in cutaneous epithelial cells [83,98,99,100,101,102,103]. The cutaneous HPV E6 proteins interact with MAML1 instead of E6AP which is the ubiquitin ligase that is utilized by mucosal high-risk HPV E6 proteins to target p53 for degradation [104]. Inhibition of TGF-β and NOTCH signaling by E6 inhibits differentiation of squamous epithelial cells [98,99,101,105]. Additionally, the cutaneous HPV E6 proteins also thwart the DNA damage response and increase the susceptibility of the infected cells to UV-induced DNA damage at least in part through interactions with the p300 protein [106,107,108,109]. While cutaneous HPV E6 proteins do not detectably interact with E6AP and thus do not target p53 for degradation, a study that investigated the effects of β-HPV E6 proteins on p53 reported that a subset of cutaneous HPV E6 proteins may inhibit activation of p53 by DNA damage [110]. These activities could all contribute to how some cutaneous HPV E6s contribute to tumorigenesis. While a number of cutaneous HPV E7 proteins bind pRB through LXCXE motifs, they do not promote pRB degradation and a subset of cutaneous γ-HPV E7 proteins lack an LXCXE motif but rather bind pRB through their C-terminal sequences [30,111,112]. The association of cutaneous HPVs with skin cancers remains limited to EV patients and, possibly, also long-term immunosuppressed individuals. However, it must be pointed out that viral sequences are not expressed in every tumor cell. Nevertheless, the E6 and E7 proteins of some cutaneous β-HPVs do have pro-tumorigenic activities that may help promote development of cancer if only through a hit and run mechanism in humans. 

## 4. Animal Models for Studying Oncogenic Activities of PVs

A key tool for studying disease is the utilization of a pre-clinical small animal model to translate findings from tissue culture-based studies to in vivo studies. Small animal models provide the opportunity to investigate key activities of oncogenic virus activities in a physiologically relevant setting and to test therapeutics. However, as pointed out earlier, the species specificity of HPVs has been a strong barrier in the ability to study viral pathogenesis. Some animal models, however, have been available to circumvent this barrier.

### 4.1. Transgenic Animal Models

A major accomplishment in the papillomavirus field was the development of HPV transgenic mouse models wherein HPV genes are expressed in epithelial cells from cellular promoters such as the human keratin K14 promoter [113,114]. Such studies have confirmed the importance of the papillomaviral E6 and E7 proteins in promoting carcinogenesis in vivo through activities including their degradation of p53 and inhibition of pRB [115,116,117,118,119,120,121,122]. Beyond this, these models have enabled our understanding of how other biological activities of these viral proteins may also promote disease [120,123]. For example, studies where mice were engineered to lack pRB and the related p107 and p130 pocket proteins developed high-grade dysplastic disease with a non-significant fraction developing cervical carcinoma in contrast to nearly all HPV16 E7 transgenic mice developing cervical carcinomas [124]. This study provides further evidence that high-risk HPV E7 proteins have oncogenic activities in addition to the inactivation of pRB and related p107 and p130 proteins [124]. HPV 16 E5 transgenic animals developed phenotypes in the skin in particular thickening of the ear skin with severity dependent on expression of the E5 transgene [125]. A subset of HPV16 E5 transgenic mice do exhibit an expansion of BrdU positive cells in the suprabasal compartment (which is dependent on number of copies of the E5 transgene) [125]. HPV 16 E5 transgenic mice are susceptible to spontaneous skin tumor development in the majority of animals [125,126]. HPV 16 E5′s enhancement of EGFR signaling contributes to these phenotypes as mice with a mutant EGFR, EGFRWa5, which acts as a dominant negative mutant of EGFR, had a reduction in E5-induced phenotypes suggesting that EGF signaling is important for HPV 16 E5′s carcinogenic effects [125,126]. These models have not only shed light on the oncogenic properties of the viral E5, E6, and E7 oncogenes but have also revealed how these viral oncogenes collaborate with co-carcinogens including exogenous estrogen treatment in cervical cancer, topical treatment with DMBA in anal cancer, and 4-NQO treatment through drinking water (mimics DNA damage induced by smoking) in head and neck cancer [127,128]. Treatment of HPV16 transgenic mice with these co-carcinogens synergized with HPV oncogene expression to promote high-grade dysplasia and cancer [127,128,129]. In addition, HPV16 transgenic mice also enabled scientists to understand how cellular mutations in key signaling pathways synergize with viral oncogene activity to promote malignant disease including PI3K and NOTCH signaling [130,131,132]. In summary, transgenic animal models have provided important insight into biological mechanisms of HPV-induced cancer. 

Transgenic animal models for the cutaneous HPV8 and HPV38 associated with cancer in EV patients have documented the carcinogenic activities of these HPVs [21,22,23,24,31,32,33,34]. Transgenic mice that express the complete early region of HPV8 expressed from the K14 promoter spontaneously develop papillomas at a high frequency that progressed to cutaneous squamous cell carcinoma (cSCC) in a subset of these mice [21]. This study clearly demonstrated that the EV-associated, cutaneous HPV8 has carcinogenic potential [21]. These mice developed disease at a faster rate when treated with UV irradiation which mimics observations in human EV patients that preferentially develop cSCC on sun exposed skin [34]. The E6 and E7 oncogenes contribute to the oncogenic phenotypes as HPV8 E6 and E6/E7 expressing mice where expression is driven by the K14 promoter are susceptible to spontaneous papilloma development and cSCC phenotypes which were enhanced by UV-treated [21,33,34]. In addition, HPV38 E6 and E7 transgenic mice are highly susceptible to cSCC development when treated with UV or chemical carcinogens [22,23,24]. A key study revealed the importance of the HPV38 E6 and E7 oncogenes in initiation of HPV-induced skin cancer and a “hit-and-run” mechanism for cutaneous HPV-induced cancer [22]. Conditional HPV38 E6/E7 transgenic mice were generated wherein the transgenes were flanked by Lox sites and were treated with UV to induced skin neoplasia [22]. At 22–25 weeks post-treatment, lesions on mice were electroporated to deliver a Cre expressing plasmid to UV-induced lesions, and they were continued to be treated with UV until 30 weeks [22]. The authors found that HPV38 E6/E7 expression was significantly decreased yet the lesions continued to progress to cSCC suggesting that the viral oncogenes are required for initiation of pre-malignant lesions but not required for the survival of the cSCC [22]. The UV-induced lesions in the HPV38 E6/E7 transgenic mice showed elevated mutational burden characteristic of UV-induced DNA damage and mutations in genes associated with cSCC development were observed including mutations in p53 and NOTCH [22]. This is particularly interesting because in cSCCs in EV patients contain less than a single copy of viral genome per cell suggesting that the virus is not required for persistence of cSCC, but, rather, may contribute to early stages of cancer development and this experiment provided mechanistic evidence for a hit-and-run mechanism by which cutaneous HPVs drive malignant disease but are not necessary for maintenance of the cancers [22]. These studies provide clear evidence that certain cutaneous β-HPVs have carcinogenic potential, and that their inhibition of tumor suppressive activities likely contribute to these phenotypes. As these models continue to expand our understanding of cutaneous HPV-induced cancer, the continued development of physiologically relevant animal models that enable the study of HPV-induced cancer at a various stages of disease are needed.

### 4.2. Natural Infection Models

As pointed out in the previous section, transgenic mouse models have been powerful tools in our understanding of the biological activities of the HPV oncoproteins proteins. In most of the models, however, the viral genes are ectopically expressed from a heterologous promoter and in every cell of a given tissue [133]. This state of constant expression more closely replicates the later stages of HPV infected cells progressing to cancer and mimics integration and fails to recapitulate the earlier events that happen after the initial infection and during viral replication. These shortcomings can be remediated with natural infection models where papillomaviruses are used to infect their natural host. Several natural infection models have been developed including Bovine (BPV), cotton-tail rabbit (CRPV) and multimammate (MnPV) papillomaviruses. Other notable animal papillomaviruses that have been invaluable include the non-human primate papillomaviruses and canine oral papillomavirus. The non-human primate papillomaviruses cause mucosal disease and cancer similar to humans and have been used to study sexual transmission of PVs [134,135,136,137,138]. Canine oral papillomavirus along with CRPV provided key insights into vaccine development and the protection provided by vaccines in preventing PV-induced disease and cancer [139,140,141,142,143,144,145,146]. BPV has been used to experimentally infect cattle where it causes large productive lesions, fibropapillomas, but it is not feasible to use BPV1 for genetic studies in vivo [147,148]. Early studies of BPV1 using in vitro based models have provided insight into the biology of various PV encoded genes but these model systems do not recapitulate the full life cycle of the papillomavirus [149,150,151,152,153,154,155,156,157,158,159,160,161]. Rodent papillomaviruses have played a critical role in our understanding of PV-induced disease with notable members including MnPV1 and McPV2 [162,163,164]. The MnPV1 infection-based model system causes benign skin papillomas in multimammate mice [162,163,165]. Multimammate mice infected with MnPV1 develop severe neoplastic disease and malignant disease when treated with co-carcinogens DMBA and TPA (two-stage carcinogenesis model) providing evidence of the pro-tumorigenic potential of PVs [166]. Transgenic animal models that express the MnPV1 E6 gene do not overtly cause neoplastic skin disease but treatment using the two-stage carcinogenesis model (DMBA and TPA treatment) leads to the development squamous cell carcinoma providing evidence for the pro-tumorigenic activities of virally encoded genes such as E6 [167]. The CRPV model system has shed light on a number of PV-induced diseases including cutaneous and mucosal diseases. Additionally, experimental infections with CRPV mutants have provided interesting insights into the role of the E6, E7, and E4 genes in viral pathogenesis [89,168,169,170,171]. However, genetic manipulation of rabbits is limited and maintaining rabbits is expensive and hence this model has only seen limited use. The limitations of the existing animal models have left an unmet challenge in the study of PV-induced disease.

### 4.3. MmuPV1 Infection-Based Mouse Model System: A New Era for PV Research

In 2011, a naturally occurring papillomavirus was discovered in a colony of nude mice in India [36]. This murine papillomavirus, now termed MmuPV1, was fully sequenced and has been used to infect both immunocompetent and immunocompromised mice [172,173,174,175,176,177,178,179]. Experiments can be done by infecting scarified skin with in vitro generated quasiviruses, by applying recircularized DNA genomes MmuPV1 or by using virus generated from infected mice to experimentally infect tissues. Through these means one can establish persistent viral infection that causes neoplastic disease and cancer in cutaneous and mucosal tissues that recapitulate features of HPV-induced disease [98,172,174,175,176,177,178,179]. Mice infected with MmuPV1 support the entire viral replication cycle in both cutaneous and mucosal tissues wherein virus can be naturally transmitted in immunocompromised animals, leading to the development of disease at new sites within the same animal [180]. Studies with MmuPV1 have been performed that examine a number of traits of PV infection including infection and transmission including sexual transmission [175]. With MmuPV1′s ability to infect laboratory mice, the field has gained a powerful new tool with which to perform studies that examine biological alterations that occur during PV infection by performing genetic studies at the viral and host level.

## 5. Biological Activities of the MmuPV1 E6 and E7 Proteins

Exciting studies examining the biological activities of MmuPV1 E6 and E7 are shedding light into how specific biochemical activities of the E6 and E7 proteins contribute to neoplastic disease induced by MmuPV1. MmuPV1 falls within the Pi genus, which contains other rodent PVs and is more closely related to the cutaneous β- and γ-HPVs than the mucosal α-HPVs [181]. These similarities can be observed in the amino acid sequence, as MmuPV1 E6 and E7 share a number of amino acids with cutaneous HPVs only (Figure 1 and Figure 2). Recent studies have revealed that the MmuPV1 E6 and E7 proteins are each required for viral pathogenesis [98,182]. Ongoing research efforts are identifying cellular targets and effectors of the MmuPV1 E6 and E7 proteins that are helping us relate this animal papillomavirus model to HPV-induced disease.

### 5.1. Known Activities of the MmuPV1 E6 Protein

Affinity purification coupled to proteomic analyses of MmuPV1 E6 in human cells demonstrated that MmuPV1 E6 protein binds MAML1, a key co-activator of NOTCH signaling, as well as the SMAD2/SMAD3 transcription factors that mediate the transcriptional output of TFG-β signaling (Figure 3). Of note, the NOTCH and the TGF-β pathways are tumor suppressive in keratinocytes [83,98,100,102,103,130]. MmuPV1 E6′s interaction with SMAD2/SMAD3 impairs TGF-β signaling by disrupting SMAD2/SMAD3′s interaction with SMAD4, leading to repression of transcriptional activation of TGF-β responsive genes including p15^INK4B^ [98]. Similarly, MmuPV1 E6′s interaction with MAML1 dampens NOTCH signaling [98]. The main biological consequence of MmuPV1 E6′s inhibition of TGF-β and NOTCH signaling is the inhibition of epithelial differentiation as evidenced by a marked delay in differentiation and increased cell survival of keratinocytes following experimental induction of keratinocyte differentiation by high calcium and serum treatment [98]. Consistent with this finding, it has been shown that MmuPV1 E6′s interaction with MAML1 and inhibition of NOTCH signaling promotes basal cell identity [183]. These authors showed that in confluent 2D cell culture studies, the MmuPV1 E6 expressing cells were able to continue proliferating following confluency and lost their response to contact inhibition of cell growth that correlated with lower levels of expression of the Keratin K10 differentiation marker [183]. Experimental infection of FoxN/N1 nude mice with MmuPV1 genomes expressing a MAML1-binding defective E6 mutant, E6^R130A^, did not induce skin disease [98]. Additionally, the MmuPV1 E6^R130A^ mutant does not impair NOTCH signaling nor the ability of MmuPV1 E6 to inhibit keratinocyte differentiation [98,183]. Therefore, MmuPV1 E6′s inhibition of differentiation and promotion of basal identity likely contribute to MmuPV1 pathogenesis. The MmuPV1 E6^R130A^ mutant may also affect interactions with other LXXLL motif containing cellular proteins including the IRF3 transcription factor, paxillin (PXN) and the E6AP (UBE3A) ubiquitin ligase. Therefore, it cannot be ruled out that loss of interaction with such proteins may also contribute to the observed phenotype indicating that additional studies are required [98]. Moreover, it is unclear how other activities of MmuPV1 E6, such as inhibition of TGF-β signaling contribute to viral pathogenesis.

### 5.2. Known Activities of MmuPV1 E7

Despite lacking an LXCXE motif which is known to mediate the ability of many HPV E7 proteins to bind the tumor suppressor, pRB, affinity purification/proteomic analyses of MmuPV1 E7 in human cells revealed that MmuPV1 E7 can interact with pRB [6,111,112,182] (Figure 4). Interestingly, and unlike LXCXE domain containing papillomavirus E7 proteins, the interaction of MmuPV1 E7 with pRB does not promote E2F transcription factor activity [182]. However, an intact pRB binding site on MmuPV1 E7 is important for viral pathogenesis and experimental infection with an MmuPV1 genome expressing the pRB binding defective E7 mutant, E7^D90A^, markedly inhibited viral pathogenesis in that papillomas arose later and were significantly smaller [182]. The exact impact of MmuPV1 E7′s interaction with pRB remains enigmatic. Unlike LXCXE domain containing E7 proteins, MmuPV1 E7 interacts with sequences in pRB’s C-terminal domain [182]. A number of “non-canonical” activities of pRB have been mapped to its C-terminus and it is currently unknown which, if any of these non-canonical activities, are abrogated or inhibited by MmuPV1 E7′s binding pRB (Figure 4). It is also important to note that MmuPV1 E7′s interaction with pRB is unlikely to account for all the activities of MmuPV1 E7 that promote pathogenesis because E7-null MmuPV1 does not cause any disease, unlike E7^D90A^ mutant MmuPV1 [182]. 

Proteomic analyses of MmuPV1 E7-associated cellular proteins in human cells also identified several other candidate cellular targets of MmuPV1 E7. One that is of particular interest is the non-receptor tyrosine phosphatase PTPN14 that is a known binding partner of many HPV E7s, including high-risk mucosal HPV E7s [80]. Determining if MmuPV1 E7 interacts with PTPN14 and if this interaction plays a role in viral pathogenesis using a mutant MmuPV1 E7 that is defective in this interaction will provide important insights into the role of a common cellular target of HPV E7 proteins [80] (Figure 4). If a mutant MmuPV1 E7 defective in its interaction with PTPN14 causes disease, then characterizing the phenotypes of lesions induced by this mutant virus will shed light on the potential biological role of this interaction. Studies are necessary to determine if MmuPV1 E7′s interaction with PTPN14 alters keratinocyte differentiation and YAP activation since studies in cell culture models have mapped alterations in these cellular processes by high-risk HPV E7 proteins to its interaction with PTPN14 and/or PTPN14′s degradation [76,77,78,79]. Further studies are needed to validate other interactions of MmuPV1 with cellular proteins and to determine if they may also contribute to viral pathogenesis.

Collectively the research on MmuPV1 E6 and E7 have already provided important insights into how specific interactions of the viral E6 and E7 proteins may contribute to viral pathogenesis.

## 6. How do MmuPV1 E6 and E7 Relate to the HPV E6 and E7 Proteins?

### 6.1. Similarities between MmuPV1 and Cutaneous HPV E6 Proteins

Unlike the “high-risk” mucosal HPV E6, MmuPV1 E6 does not target p53 for degradation (Figure 5). Instead, MmuPV1 E6 targets the NOTCH and TGF-β signaling pathways similar to cutaneous HPV5 and HPV8 E6 proteins [98,99] (Figure 5). As stated above, MmuPV1′s interaction with LXXLL motifs is critical for pathogenesis [98,183]. Since these activities are shared with some cutaneous HPVs, particularly the EV-associated HPV5 and HPV8 strengthens the argument that the shared activities of the MmuPV1 E6 protein and cutaneous HPV E6 oncoproteins contribute to tumorigenesis because transgenic mice that express the HPV8 E6 oncogenes are susceptible to squamous cell carcinoma in conjunction with co-carcinogens, i.e., UV-B [21,22,23,24,33,34]. It is particularly interesting that MmuPV1 can also cause mucosal disease, including cervical lesions, similar to high-risk α-HPV [174,175,176,177,178,179]. MmuPV1 E6 interacts with MAML1 instead of E6AP and hence does not target p53 for degradation [104]. However, similar to MmuPV1 E6 which inhibits NOTCH signaling through MAML binding, high-risk HPV E6 proteins may also inhibit NOTCH signaling, albeit indirectly by causing p53 degradation [67,83,99,101,105,184,185]. Similarly, high-risk α-HPVs also inhibit TGF-β signaling although through a mechanism that likely differs from that mediated by MmuPV1 because it is mediated by the HPV E7 protein [81,186]. Therefore, the current evidence clearly suggests that MmuPV1 E6 targets similar cellular signaling pathways as the cutaneous β-HPV 5 and 8. Moreover, the α-HPV E6 and E7 proteins may subvert these same pathways, albeit through different mechanisms and cellular targets. Additional experiments will be necessary to solidify these observations.

### 6.2. Cellular Targets and Effectors of MmuPV1 E7—Insights for HPV Pathogenesis

As with MmuPV1 E6, MmuPV1 E7 has been shown to interact with at least two known targets of HPV E7 oncoproteins, the pRB and PTPN14 tumor suppressors [6,76,77,182] (Figure 5). Given that MmuPV1 E7′s interaction does not inhibit pRB’s inhibition of E2F activity, MmuPV1 E7 may inhibit some of the “non-canonical” activities of pRB that are mediated through the pRB C-terminal domain that MmuPV1 E7 interacts with [187]. There are a number of “non-canonical” activities of pRB and one that is of particular interest is mediated by a cell cycle independent interaction of pRB with an E2F1-EZH2 complex [188]. The amino acids of pRB that are key for this interaction have been mapped to the pRB C-terminus [187,188,189,190]. EZH2 is histone methyl transferase that writes repressive trimethylation marks on histone H3 (H3K27me3). The pRB/E2F1-EZH2 complex has been shown to maintain the transcriptional repressive histone H3K27me3 modifications on repetitive sequences in the genome [188]. A pRB mutant, pRB^S^, that is defective for E2F1-EZH2 complex binding has been shown to render mice susceptible to the development of cancer, specifically lymphomas and hepatocellular carcinoma [188]. Considering that loss of this interaction makes some cell types more susceptible to developing cancer, it will be interesting to determine if MmuPV1 E7 binding may interfere with this tumor suppressive activity. There are other non-canonical activities of pRB that could play a role in MmuPV1 E7′s interaction with pRB including pRB’s interactions with ABL1 nonreceptor tyrosine kinase and F-box protein SKP2 [191,192]. A new study that examined how pRB binding sites are distributed across the genome and what transcription factors interact with pRB outside E2F proteins showed that pRB binds to AP1 and CTCF sites [193]. It will be interesting to determine whether MmuPV1 E7 affects pRB binding to these sites.

Many of the non-canonical pRB activities remain poorly studied and whether and how they may affect keratinocyte biology remains largely unknown. Additionally, pRB has also been implicated in the maintaining genome stability, chromosome condensation, and DNA damage repair (non-homologous end joining and homologous repair) [189,190,194,195,196,197]. It is important to note that, unlike low-risk HPV E7s, the high-risk HPV E7s have been shown to cause pRB degradation, which results in the abrogation of canonical as well as non-canonical pRB activities [62,63,64,198]. Hence, inactivation of non-canonical pRB activities may be key to the carcinogenic activities of high-risk HPVs. MmuPV1 E7 may provide an excellent tool to identify these activities in mechanistic detail.

As discussed previously, the high-risk HPV E7 proteins have pRB independent activities that play a role in promoting tumorigenesis as mutant forms of E7 that cannot interact with pocket proteins can still transform tissue culture cells and cause neoplastic disease in transgenic animals [120,123,199]. Of note, experimental infection of rabbits with a CRPV genome expressing a pRB defective E7 mutant, caused papillomas but at a lower frequency than WT CRPV (~12% in WT and ~3% in pRB defective binding mutant) [168]. While MmuPV1 E7′s interaction with pRB is important for viral pathogenesis, other activities contribute to E7′s ability to promote pathogenesis [168]. Of particular interest is determining if MmuPV1 E7′s interaction with PTPN14 plays a role in viral pathogenesis and examining the phenotypes that arise using a mutant MmuPV1 E7 defective for this interaction. Characterizing phenotypes dependent upon MmuPV1 E7′s interaction with PTPN14 could potentially provide novel insight into how this interaction perturbs the biology of keratinocytes including differentiation, promotion of basal cell identity, and potentially other biological outcomes of inhibiting HIPPO signaling that may also be altered by the high-risk HPV E7 proteins [76,78,79,80].

## 7. Conclusions and Future Directions

The MmuPV1 mouse model has already provided important insights into the biological activities of E6 and E7 which are relevant for HPV pathogenesis and potentially carcinogenesis. Future work is still needed to better understand the interplay between interactions with cellular targets and effectors, in particular understanding how these interactions promote proliferation and other aspects of neoplastic disease outside of inhibiting differentiation. The development of transgenic animal models that express the MmuPV1 E6 and E7 oncoproteins have the potential to determine if these viral oncogenes are carcinogenic on their own or require co-factors, i.e., estrogen, UV, or others, to drive cancer development in mice. Additional studies can also be performed to determine how other early viral proteins contribute, including the viral transcriptional activator E2, to these phenotypes. An outstanding question is which activities of the MmuPV1 E6 and E7 oncogenes promote proliferation and increases E2F activity since MmuPV1 E7 clearly does not perform this activity through its interaction with pRB [182]. Continued work determining how the different activities of MmuPV1 E6 and E7 contribute to viral pathogenesis during a natural infection using viral mutants will be informative, in particular examining activities that are shared with HPV E6 and E7 oncogenes (both cutaneous and high-risk) which may shed light on species specificity of papillomaviruses. MmuPV1′s ability to infect immunocompetent animals, also provides a rich opportunity to combine natural infection with transgenic or knock-out mice to explore specific MmuPV1 E6 and E7 cellular targets or effectors that contribute to viral pathogenesis or immune evasion. These complementation studies would be invaluable to understanding the mechanisms of the viral persistence and carcinogenesis. A prime example is determining if MmuPV1 E6′s interaction with MAML1 is the only LXXLL motif containing cellular protein that contribute to the phenotype observed in the E6^R130A^ mutant (the LXXLL motif binding mutant) [98]. Utilizing a transgenic mouse model that expresses a dominant negative MAML1 (dnMAML1) that inhibits NOTCH activity would clearly address this question (these animals have been used to study HPV and NOTCH signaling in vivo) [130]. A lack or partial complementation of dnMAML1 infected mice with the E6^R130A^ mutant would suggest that other LXXLL binding partners of MmuPV1 E6 contribute to viral pathogenesis and a complete complementation would suggest binding of MAML1 through an LXXLL motif, which inhibits NOTCH signaling, is the only activity that contributes to viral pathogenesis. Similar studies can also be performed by infecting mice that have been genetically alter TGF-β signaling to clearly determine the importance of inhibiting TGF-β signaling for viral pathogenesis. Similar complementation studies can be performed to understand the importance of specific biological activities of MmuPV1 E7 including its interaction with pRB. Utilizing animals that encode an pRB mutant that lacks non-canonical activities of pRB, i.e., pRB^S^ mutant mice, infection with the RB-binding defective E7^D90A^ mutant MmuPV1 could reveal whether loss of non-canonical activities of pRB complement for the loss of pRB binding and provide evidence on which non-canonical activities of pRB play a role in MmuPV1 pathogenesis [188]. Studies on MmuPV1 E6 and E7 will continue to provide important mechanistic insights how these viral proteins contribute to pathogenesis and these insights will be key to illuminating the oncogenic activities of cutaneous and mucosal HPVs.

## Figures and Tables

**Figure 1 viruses-14-02138-f001:**
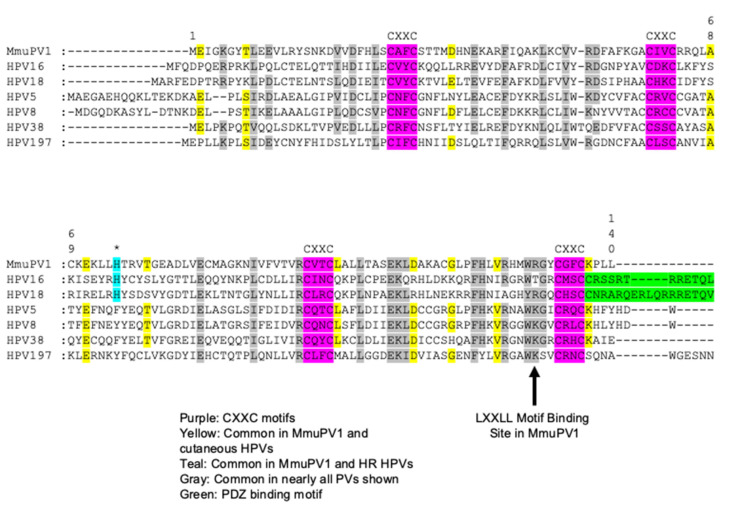
**Alignment of MmuPV1 E6 protein to cancer associated HPV E6 proteins.** Amino acid sequence alignment of the MmuPV1 E6 protein with E6 proteins encoded by notable cancer associated HPVs including the high-risk α-HPV16 and HPV18, the EV-associated HPVs 5, 8, and 38, and the cutaneous γ-HPV-197. The positions of the four C-X-X-C (C, cysteine; X, any amino acid) motifs which bind Zn^2+^ amino acid sequence in MmuPV1 E6 are conserved with the HPV E6 proteins (highlighted in purple). MmuPV1 E6 does not contain the PDZ binding domain present in the high-risk mucosal HPV E6 proteins (highlighted in green). A number of amino acids are conserved between MmuPV1 and all the HPV genomes (highlighted in grey). A number of amino acids are conserved between the cutaneous HPV E6 proteins and MmuPV1 E6 as predicted as MmuPV1 is phylogenetically more closely related to the cutaneous HPVs (highlighted in yellow). There was a single instance where an amino acid in MmuPV1 E6 was exclusively shared with the high-risk mucosal HPV E6 proteins (highlighted in teal and asterisk above). A critical amino acid for LXXLL motif binding in MmuPV1 is indicated with the arrow.

**Figure 2 viruses-14-02138-f002:**
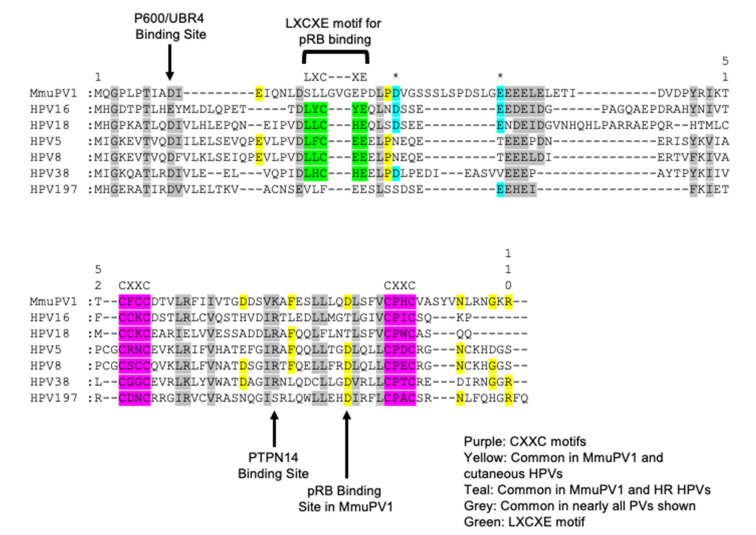
**Alignment of the MmuPV1 E7 protein to cancer associated HPV E7 proteins.** An amino acid sequence alignment was performed comparing the MmuPV1 E7 protein to various cancer associated HPV genotypes including the high-risk HPV 16 and 18, EV-associated β-HPV 5, 8, and 38, and the cutaneous γ-HPV 197. Like the caner associated HPV E7 proteins, MmuPV1 E7 contains two CXXC zinc binding motifs present in the C-terminal domain (highlighted in purple). MmuPV1 E7 does not contain an LXCXE motif like the high-risk and cutaneous β-HPVs used in this alignment (highlighted in green). A number of amino acids are shared between the various papillomavirus E7 proteins in particular in between the C-X-X-C motifs (highlighted in grey). As expected, a number of amino acids are shared with the cutaneous HPVs used in this alignment (highlighted in yellow). Interestingly, there were a couple of amino acids that were present in MmuPV1 that were more likely to be present in the high-risk HPV E7 oncogenes compared to the cutaneous HPV E7 oncogenes (highlighted in teal and asterisk above). This alignment again provides some evidence that MmuPV1 E7 is more closely related to the cutaneous HPV genotypes than the high-risk HPV genotypes. A few notable binding sites for E7 are noted with arrows including the amino acids responsible for binding p600 (UBR4) and PTPN14 and the pRB binding site in MmuPV1.

**Figure 3 viruses-14-02138-f003:**
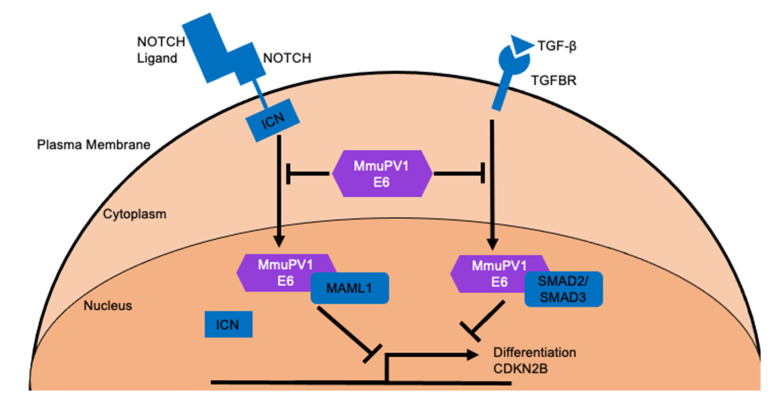
**MmuPV1 E6 cellular targets and the outcomes of these interactions.** MmuPV1 E6 has been shown to interact with MAML1 and SMAD2/SMAD3 which are critical co-activators of NOTCH and TGF-β signaling, respectively [98]. MmuPV1 E6 binds to these important transcriptional activators and prevents their ability to promote transcriptional activation of NOTCH and TGF-β responsive genes following stimulation with appropriate ligand. One important biological outcome of these interactions is the inhibition of keratinocyte differentiation [98].

**Figure 4 viruses-14-02138-f004:**
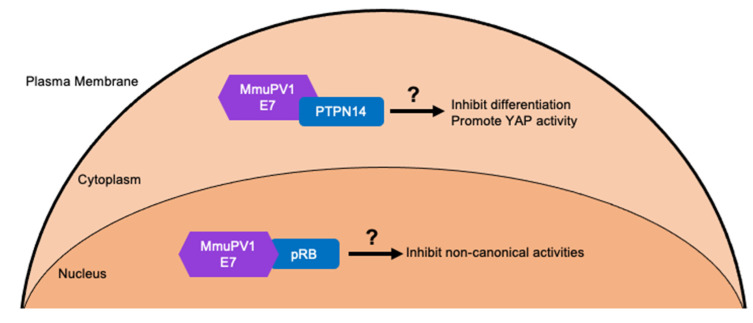
**MmuPV1 E7 cellular targets and predicted outcomes.** MmuPV1 E7 has been shown to interact with the tumor suppressor pRB [182]. However, the activities of pRB that are affected by this interaction remain unknown. The prediction is that MmuPV1 E7 likely affects non-canonical activities of pRB which are activities not associated with cell cycle regulation since MmuPV1 E7 does not increase E2F activity. Through affinity purified/ mass spectrometry analysis several other cellular targets of MmuPV1 E7 have been identified including the cellular non-receptor tyrosine phosphatase PTPN14 which is a common target of HPV E7 proteins [182]. MmuPV1 E7′s interaction with PTPN14 may inhibit keratinocyte differentiation and/or promote basal cell identity by increasing YAP activity. This is predicted from studies with high-risk HPV E7 proteins that bind and destabilize PTPN14 [78,79].

**Figure 5 viruses-14-02138-f005:**
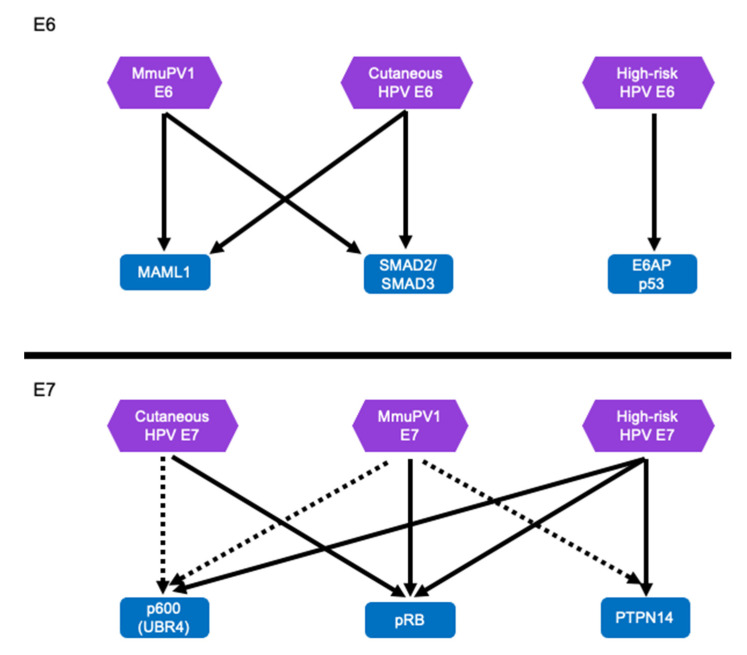
**MmuPV1 E6 and E7 and HPV E6 and E7 proteins appear to target similar cellular proteins and pathways.** The MmuPV1 E6 protein targets MAML1 and SMAD2/SMAD3 to inhibit NOTCH and TGF-β signaling, respectively. The cutaneous β-HPV E6 protein target the same cellular proteins leading to similar cellular phenotypes including inhibition of keratinocyte differentiation. This is distinctly different from the high-risk HPV E6 protein which targets p53 for degradation. However, the outcome of p53 degradation by the high-risk HPV E6 may also cause inhibition of NOTCH signaling [66,67,83,185]. The MmuPV1 E7 and the HPV E7 proteins likely target similar cellular binding partners including pRB to promote efficient neoplastic disease. It is likely that MmuPV1 E7 protein also target several other well-known HPV E7 cellular targets including p600 (UBR4) and PTPN14. Well validated cellular interacting proteins are indicated with arrows, interactions tentatively identified by proteomic analyses are show indicated with dashed lines.

## Data Availability

Not applicable.

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
