# Peer review of "Molecular Mechanisms of MmuPV1 E6 and E7 and Implications for Human Disease"

_viruses, 2022, doi:10.3390/v14102138_

Round 1
Reviewer 1 Report
The review by Romero-Masters and colleagues provides a useful overview of the actions of the E6 and E7 proteins of mouse papillomavirus, MmuPV1 in comparison to those of high-risk HPVs along with other PVs. The review is thorough and is easy to read in most parts except as noted below. Several points can be addressed to improve the readability of the manuscript.
1). The two sections describing the activities of MmuPV1 E6 (340-369) and (424-449) are fairly repetitive and can be either combined into one or the repeated statements pruned.
Minor:
1). Line 115 fix” decrease degrade”
2). Line 131: high risk E6 and E7 proteins have been shown to activate and increase levels of DNA repair factors not inhibit them
3). Line 140: only a few b-HPVs can immortalize cells not all
4). Lines 183-190. : run on sentence
5). Lines 193-197: run on sentence
6). Lines 412-416: run on sentence
Author Response
Major Comment
Comment: The two sections describing the activities of MmuPV1 E6 (340-369) and (424-449) are fairly repetitive and can be either combined into one or the repeated statements pruned.
Response: We have pruned the sections to remove redundant statements and focus the sections (See lines 345-374 and 434-460).
Minor comments
Comment 1: Line 115 fix” decrease degrade”
Response: This has been fixed (see line 116).
Comment 2: Line 131: high risk E6 and E7 proteins have been shown to activate and increase levels of DNA repair factors not inhibit them.
Response: We have corrected this statement (see line 133).
Comment 3: Line 140: only a few b-HPVs can immortalize cells not all
Response: We have corrected this statement to be more accurate (see lines 142-143).
Comment 4: Lines 183-190: run on sentence.
Response: We have changed this sentence (see lines 185-191).
Comment 5: Lines 193-197: run on sentence.
Response: We have changed this sentence (see lines 195-200).
Comment 6: Lines 412-416: run on sentence.
Response: We have corrected this sentence (see lines 418-424

Reviewer 2 Report
This is an extensive review article on the activities of MmuPV1 E6 and E7 proteins. I have only some minor comments
1. What do you think drives the species specificity of PV infections? Is it the LCR or is it the different E6/E7 activities?
2. The interacting partners of MmuPV1 E6/E7 proteins identified by proteomics analysis; were these experiments carried in mouse keratinoytes? If not, do you think the interactome might be different in mice vs human?
3. E2 is the master activator/repressor of the viral early promoter. The tumorigenesis may be driven by E6/E7 but E2 might play a role in the viral lifecycle. the diverse activities of E2 may also be important in this aspect.
4. Even though there are transgenic HPV E6/E7 models driven by K14 promoter, is there a need for a transgenic model with MmuPV1 E6/E7? Will it develop tumor without a cofactor?
5. Line 115; PTPN14 is not a kinase
6. Spell check line 328,537
Author Response
Minor Comments:
Comment 1: What do you think drives the species specificity of PV infections? Is it the LCR or is it the different E6/E7 activities?
Response: This is an interesting question, and a comment has been added to the conclusion about how some of the proposed studies could help address this (see line 539).
Comment 2: The interacting partners of MmuPV1 E6/E7 proteins identified by proteomics analysis; were these experiments carried in mouse keratinoytes? If not, do you think the interactome might be different in mice vs human?
Response: The published data is in human cells, and this has been added to the text (see lines 345, 385, and 411).
Comment 3: E2 is the master activator/repressor of the viral early promoter. The tumorigenesis may be driven by E6/E7 but E2 might play a role in the viral lifecycle. the diverse activities of E2 may also be important in this aspect.
Response: We have added comment in the conclusion that future studies can be performed to address the role of other early proteins and their impact on pathogenesis (see lines 531-533).
Comment 5: Even though there are transgenic HPV E6/E7 models driven by K14 promoter, is there a need for a transgenic model with MmuPV1 E6/E7? Will it develop tumor without a cofactor?
Response: This is an interesting idea, and a comment has been added to the conclusion on how these models may be used to answer this question (see lines 528-531).
Comment 5: Line 115; PTPN14 is not a kinase.
Response: We have corrected this to phosphatase. Thanks for catching this mistake (see line 117).
Comment 6: Spell check line 328,537.
Response: We have made corrections to the incorrect spellings (see line 329 and 547).
